# Novel Mn^4+^-Activated K_2_Nb_1−*x*_Mo*_x_*F_7_ (0 ≤ *x* ≤ 0.15) Solid Solution Red Phosphors with Superior Moisture Resistance and Good Thermal Stability

**DOI:** 10.3390/molecules28114566

**Published:** 2023-06-05

**Authors:** Yuhan Gao, Lei Feng, Linglin Wang, Jun Zheng, Feiyao Ren, Siyu Liu, Zhanglei Ning, Ting Zhou, Xiaochun Wu, Xin Lai, Daojiang Gao

**Affiliations:** College of Chemistry and Materials Science, Sichuan Normal University, Chengdu 610066, China; 2020080608@stu.sicnu.edu.cn (Y.G.); 20201201055@stu.sicnu.edu.cn (L.F.); 20221201058@stu.sicnu.edu.cn (L.W.); 2020080243@stu.sicnu.edu.cn (J.Z.); 2020080221@stu.sicnu.edu.cn (F.R.); 2020080622@stu.sicnu.edu.cn (S.L.); ting_zhou@sicnu.edu.cn (T.Z.); wuxiaochun@sicnu.edu.cn (X.W.); laixin1972@sicnu.edu.cn (X.L.)

**Keywords:** fluoride, phosphor, luminescence performance, thermal stability, moisture resistance

## Abstract

Nowadays, Mn^4+^-activated fluoride red phosphors with excellent luminescence properties have triggered tremendous attentions for enhancing the performance of white light-emitting diodes (WLEDs). Nonetheless, the poor moisture resistance of these phosphors impedes their commercialization. Herein, we proposed the dual strategies of “solid solution design” and “charge compensation” to design K_2_Nb_1−*x*_Mo_*x*_F_7_ novel fluoride solid solution system, and synthesized the Mn^4+^-activated K_2_Nb_1−*x*_Mo_*x*_F_7_ (0 ≤ *x* ≤ 0.15, *x* represents the mol % of Mo^6+^ in the initial solution) red phosphors via co-precipitation method. The doping of Mo^6+^ not only significantly improve the moisture resistance of the K_2_NbF_7_: Mn^4+^ phosphor without any passivation and surface coating, but also effectively enhance the luminescence properties and thermal stability. In particular, the obtained K_2_Nb_1−*x*_Mo*_x_*F_7_: Mn^4+^ (*x* = 0.05) phosphor possesses the quantum yield of 47.22% and retains 69.95% of its initial emission intensity at 353 K. Notably, the normalized intensity of the red emission peak (627 nm) for the K_2_Nb_1−*x*_Mo_*x*_F_7_: Mn^4+^ (*x* = 0.05) phosphor is 86.37% of its initial intensity after immersion for 1440 min, prominently higher than that of the K_2_NbF_7_: Mn^4+^ phosphor. Moreover, a high-performance WLED with high CRI of 88 and low CCT of 3979 K is fabricated by combining blue chip (InGaN), yellow phosphor (Y_3_Al_5_O_12_: Ce^3+^) and the K_2_Nb_1−*x*_Mo_*x*_F_7_: Mn^4+^ (*x* = 0.05) red phosphor. Our findings convincingly demonstrate that the K_2_Nb_1−*x*_Mo_*x*_F_7_: Mn^4+^ phosphors have a good practical application in WLEDs.

## 1. Introduction

White light-emitting diodes (WLEDs) is regarded as the green lighting or the fourth generation solid-state lighting on account of brilliant efficiency, long operation lifetime, high reliability, small size, environmentally friendly, and low power consumption [1,2,3,4,5,6,7,8,9]. Recently, WLEDs have been zealously explored as the main light source for display and illumination in numerous areas including auto mobile, liquid crystal displays (LCDs), medical treatment, agriculture and so on [10,11,12,13,14,15]. Currently, the most commercially popular approach for fabricating WLEDs is an integration of blue diode chips (InGaN) and yellow Y_3_Al_5_O_12_: Ce^3+^ (YAG: Ce^3+^) phosphor [16,17,18]. Nevertheless, YAG: Ce^3+^-based WLEDs faces two severe shortcomings of high correlated color temperature (CCT > 4500 K) and low color rendering index (CRI < 80), which is the result of notoriously insufficient red contribution, thus restricting their wider commercial application [1,10,19,20]. Interestingly, adding suitable red phosphor to supplement the scarcity of red emission could overcome the bottleneck mentioned above [1]. As a typical non-rare earth activator of the red emission, Mn^4+^ ion has received vast attention due to its characteristics of wide absorption, strong and sharp emission, low reabsorption, easily available and low cost. At the same time, Mn^4+^-activated oxides and fluorides are developing into the two categories of red phosphors with a great prospect [21]. Since the covalent bond of Mn^4+^‒O^2−^ in oxide is stronger distinctly than that of Mn^4+^‒F^−^ in fluoride, so the emission wavelength of Mn^4+^ in oxide is too long to be sensitive to human eyes. Furthermore, the strongest excitation wavelength of Mn^4+^-activated fluoride is largely matching the emission peak of blue light LED chip and it is easily perceived by the human eye because the emission peak falls in the red light region, together with good chemical stability, high efficiency and simple synthesis method, making it be an ideal red phosphor for WLEDs [20].

To our knowledge, broad absorption and narrow-band emission at about 630 nm is the remarkable characteristics of Mn^4+^-activated fluoride red phosphors owing to particular 3d^3^ outer electron configuration of Mn^4+^ [22,23,24,25]. According to the recent years reports, the Mn^4+^-activated fluoride can be divided into three types:(1) A_3_X(Ⅲ)F_6_: Mn^4+^ (A = Li, Na, K, Cs; X = Al, Ga, Sc), A_2_BX(Ⅲ)F_6_: Mn^4+^ (A/B = Li, Na, K, Rb, Cs; X = Al, Ga, Sc) [26,27,28]; (2) A_2_M(Ⅳ)F_6_: Mn^4+^ (A = NH_4_, Li, Na, K, Rb, Cs; M = Si, Ti, Ge, Sn, Zr, Hf), BM(Ⅳ)F_6_: Mn^4+^ (B = Ba, Zn; M = Si, Ge, Ti), AA′M(Ⅳ)F_6_: Mn^4+^ (A/A′ = Li, Na, K; M = Si, Ti, Ge) [29,30,31,32,33,34,35,36,37,38,39]; (3) A_2_X(Ⅴ)F_7_: Mn^4+^ (A = Li, Na, K, Rb, Cs; X = Nb, Ta) [40,41,42]. Unfortunately, the first two compounds possess symmetrical [MnF_6_]^2−^ and thereby lack the special spectral phenomenon of the zero phonon line (ZPL), which is crucial to advance the color purity quality of WLEDs [40]. Remarkably, in C_1_ (or C_2v_) group symmetry, K_2_NbF_7_: Mn^4+^ undergoes highly distorted octahedral environment, making the distinctive vibronic transitions and intense zero phonon line appear [40]. Substantially, it may dominate the emission of Mn^4+^, in favor of the ^2^E_g_→^4^A_2g_ spin-forbidden transition of Mn^4+^ and the enhancement of the color purity [43]. Hence, Mn^4+^-activated K_2_NbF_7_ phosphors have evoked plentiful attention. However, K_2_NbF_7_: Mn^4+^ phosphors can be easily hydrolyzed into manganese oxides and hydroxides in damp condition due to the inherent poor moisture resistance when [MnF_6_]^2−^ complex ions locate on the phosphor surface, leading to a reduction in the intensity of the red emission and the performance deterioration of the WLEDs during long-term operation. Therefore, it is very important and valuable to improve the moisture resistance of Mn^4+^-activated K_2_NbF_7_ red phosphors [44,45,46,47].

To solve these problems, we propose the dual strategies of “solid solution design” and “charge compensation” to design novel Mn^4+^-activated fluoride luminescent material, and the as-designed K_2_Nb_1−*x*_Mo*_x_*F_7_: Mn^4+^ red phosphors were synthesized via co-precipitation method. It should be noted that the replacement of Nb^5+^ by Mn^4+^ will probably produce an anion vacancy (fluorine vacancy) in the matrix, while the substitution of Nb^5+^ with Mo^6+^ will generate a cation vacancy and plays a role of charge compensation [42]. After doping of Mo^6+^, the structural diversity and structural rigidity enhancement of the K_2_NbF_7_: Mn^4+^ can be achieved simultaneously, thus effectively enhancing the comprehensive luminescence properties of the Mn^4+^-activated fluoride phosphors. Our results clearly demonstrated that not only the moisture resistance but also the thermal stability and luminescence performance are immensely enhanced by doping Mo^6+^. This work can provide an effective and facile strategy to design and synthesize the novel Mn^4+^-activated fluoride phosphors with superior luminescence properties, moisture resistance and good thermal stability, it also exploits new opportunities for these fluoride red phosphors used in WLEDs in outdoor high humidity working conditions.

## 2. Results and Discussion

### 2.1. Phase Structure, Morphology and Composition

Figure 1a exhibits the XRD patterns of the prepared K_2_Nb_1−*x*_Mo*_x_*F_7_: Mn^4+^ samples. Apparently, the diffraction peaks of all the samples are very consistent with the standard card of K_2_NbF_7_ (PDF#22-0839) and no impurity phase appearances, indicating that the obtained K_2_Nb_1−*x*_Mo*_x_*F_7_: Mn^4+^ phosphors are pure solid solution. Since Nb^5+^ (69 pm, CN = 7), Mo^6+^ (73 pm, CN = 7) and Mn^4+^ (53 pm, CN = 6) have similar ionic radii, they are able to occupy the same lattice site in the K_2_MF_7_ fluoride. It also can be seen that the strongest peak of the samples located at 2*θ* of ~17.5° gradually shift to the lower 2*θ* side with the increasing *x* (see in the enlarged patterns), further confirming that the replacement of Nb^5+^ by Mo^6+^ in K_2_NbF_7_ host and the formation of K_2_Nb_1−*x*_Mo*_x_*F_7_: Mn^4+^ solid solution. The crystal structure of K_2_Nb_1−*x*_Mo*_x_*F_7_: Mn^4+^ solid solution is presented in Figure 1b. It can be intuitively seen that there are seven F^−^ around Nb^5+^ and Mo^6+^, forming [NbF_7_]^2−^ decahedron and [MoF_7_]^−^ decahedron, respectively. Therefore, the defect equations of Mn^4+^ and Mo^6+^ doped into K_2_NbF_7_ host can be expressed as follows [42,48]:(1)$$K_{2}MnF_{6}\overset{K_{2}NbF_{7}}{\to }K_{K}^{\times }+Mn_{Nb}^{/}+6F_{F}^{\times }+V_{F}^{•}$$
(2)$$KMoF_{7}\overset{K_{2}NbF_{7}}{\to }K_{K}^{\times }+Mo_{Nb}^{•}+7F_{F}^{\times }+V_{K}^{/}$$

As shown in Equations (1) and (2), the replacement of Nb^5+^ by Mn^4+^ will probably produce anion vacancy (i.e., F^−^ ion vacancy), while the substitution of Nb^5+^ with Mo^6+^ will probably produce cation vacancy (i.e., K^+^ vacancy) in K_2_NbF_7_ host material. Cation vacancy can also compensate the charge of F^−^ vacancy, which is beneficial to stabilize the structure of the obtained solid solution.

The percentage difference of radius (Dr) between doped ion Mo^6+^ and replacement ion Nb^5+^ calculated by Formula (1) is 5%, which is less than 15%, indicating that Nb^5+^ and Mo^6+^ can form K_2_Nb_1−*x*_Mo*_x_*F_7_: Mn^4+^ continuous solid solution.
(3)$$\mathrm{Dr}=\frac{R_{1}(\mathrm{CN})-R_{2}(\mathrm{CN})}{R_{1}(\mathrm{CN})}\times 100\%$$
where R_1_ and R_2_ are radii of the substituted ion and matrix ion, respectively, and CN represents the coordination number.

To further explore the effect of Mo^6+^-doping concentration on the phase purity and cell parameters of the K_2_Nb_1−*x*_Mo*_x_*F_7_: Mn^4+^ samples, the Rietveld refinement was carried out based on the recorded XRD data of all samples by using GSAS software (GSAS_EXPGUI). The detailed refinement results of the K_2_Nb_1−*x*_Mo*_x_*F_7_: Mn^4+^ samples are provided in Appendix A and Table 1, respectively. As shown in Appendix A, the original XRD data (recorded data) of all the samples could well match with the corresponding calculated results based on the GSAS software, suggesting that these samples possess high phase purity. All of the K_2_Nb_1−*x*_Mo*_x_*F_7_: Mn^4+^ samples are monoclinic structure with the same space group of *P*12_1_/*c*_1_. In particular, the unit cell volume of the K_2_Nb_1−*x*_Mo*_x_*F_7_: Mn^4+^ samples gradually increased from 631.624 Å^3^ to 632.067 Å^3^ as Mo^6+^-doping concentration (*x*) increasing from 0 to 0.15, this should be attributed the cell expansion induced by the replacement of Nb^5+^ by Mo^6+^ with the larger ion radius. The obtained relatively lower reliability factors including *R_p_*, *R_wp_* and *χ^2^* for all the K_2_Nb_1−*x*_Mo*_x_*F_7_: Mn^4+^ samples indicate that the obtained refinement results are credible. The Rietveld refinement results once again confirm that the doped Mo^6+^ ions successfully replaced Nb^5+^ ions in the K_2_NbF_7_ host and formed K_2_Nb_1−*x*_Mo*_x_*F_7_: Mn^4+^ solid solution.

To investigate the influence of Mo^6+^-doping on the surface microstructure of K_2_Nb_1−*x*_Mo*_x_*F_7_: Mn^4+^ samples, as the examples, Figure 2 gives the SEM images of K_2_NbF_7_: Mn^4+^ and K_2_Nb_1−*x*_Mo*_x_*F_7_: Mn^4+^ with *x* = 0.05 and 0.15, respectively. It can be observed that all the samples are mainly rod-like microcrystals with the length between 10–70 μm and the width between 2–5 μm, representing excellent crystallinity. Additionally, all the samples have similar uniformity and regularity, suggesting that the doping concentration of Mo^6+^ has little influence on the surface microstructure of the K_2_Nb_1−*x*_Mo*_x_*F_7_: Mn^4+^ solid solution crystals. Reasonably, the influence of the surface microstructure (morphology and size) on the luminescence properties of the K_2_Nb_1−*x*_Mo*_x_*F_7_: Mn^4+^ solid solution phosphors can be ignored in the subsequent study.

The actual mole percentages of the doped Mo^6+^ and Mn^4+^ ions in K_2_Nb_1−*x*_Mo*_x_*F_7_: Mn^4+^ solid solution crystals were measured by ICP-MS, and the results are depicted in Table 2. As can be seen, the Mo^6+^-doping concentration in these phosphors gradually increases with the increasing concentration of Mo^6+^ in the starting solution, although it is much lower than that of the starting solution. It should be noted that the actual Mn^4+^-doping concentration in the phosphors is between 1.19–1.48%, basically keeping in the narrow range of 1.35±0.15%. Thus, the influence of Mn^4+^-doping concentration on the luminescence properties of the K_2_Nb_1−*x*_Mo*_x_*F_7_: Mn^4+^ phosphors also can be neglected in the following studies.

### 2.2. Luminescence Properties

The photoluminescence excitation spectra (PLE) of the K_2_Nb_1−*x*_Mo*_x_*F_7_: Mn^4+^ phosphors were recorded under the monitoring wavelength of 627 nm, as illustrated in Figure 3a. Impressively, all samples have two wide absorption peaks in the wavelength range of 300–550 nm. The peak located in the near ultraviolet (~362 nm) district is attributed to the spin permissive transition of ^4^A_2g_→^4^T_1g_ of the 3d electrons of Mn^4+^ in the octahedron, whereas the peak located in blue (~470 nm) region is derived from the spin permissive transition of ^4^A_2g_→^4^T_2g_ of the 3d electrons of Mn^4+^ in the octahedron [49,50]. The K_2_Nb_1−*x*_Mo*_x_*F_7_: Mn^4+^ phosphors have a strong absorption band in blue light region also means that these phosphors could be effectively excited by the InGaN LED blue light chip. Therefore, K_2_Nb_1−*x*_Mo*_x_*F_7_: Mn^4+^ phosphors are promising red phosphors for WLEDs, which can be expected to improve CCT and R_a_ of the packaged WLEDs [19]. Figure 3b depicts the emission spectra (PL) of the K_2_Nb_1−*x*_Mo*_x_*F_7_: Mn^4+^ phosphors in the range of 500–700 nm under excitation of 470 nm blue light. All the phosphors display the similar emission spectra, which are composed of the seven typical sharp emission peaks at 597 nm, 606 nm, 611 nm, 619 nm, 627 nm, 631 nm and 644 nm, respectively. These peaks are attributed to the anti-Stokes shift, ZPL and Stokes shift originating from ^2^E_g_→^4^A_2g_ spin-forbidden transition of Mn^4+^, respectively [43,44]. Interestingly, the intensities of these emission peaks for the K_2_Nb_1−*x*_Mo*_x_*F_7_: Mn^4+^ phosphors initially increase and then decease with the increasing Mo^6+^-doing concentration (*x*), giving the largest at *x* = 5%. It can be reasonably concluded that the best enhancement effect of luminescent properties induced by Mo^6+^-doping in K_2_NbF_7_: Mn^4+^ system should appear at *x* = 5% in our case (Figure 3b).

In general, the main factors that determine the luminescent properties of the phosphors include the doping concentration of activator, surface microstructures (morphology and size) and the local environment around the activator ions. In present study, the changes in the doping concentration of Mn^4+^ and the surface microstructures of the K_2_Nb_1−*x*_Mo*_x_*F_7_: Mn^4+^ phosphors are almost negligible (see the relevant discussion above). Thus, the variation of luminescent properties in the K_2_Nb_1−*x*_Mo*_x_*F_7_: Mn^4+^ phosphors should be attributed to the change of the local environment around Mn^4+^ in the lattices after being doped by Mo^6+^ ions. As mentioned above, the unit cell of the K_2_Nb_1−*x*_Mo*_x_*F_7_: Mn^4+^ solid solution will expand owing to the larger ion radius of Mo^6+^ than that of the Nb^5+^. Hence, the substitution of Nb^5+^ by Mo^6+^ will inevitably lead to the variation of the sublattice of the K_2_NbF_7_: Mn^4+^, further changing the local environment around Mn^4+^, thus effectively breaking through the prohibition rule to improve *d*-*d* transition probability and enhancing red emission to a certain extent.

The UV-visible diffuse reflection spectra and band gaps of K_2_NbF_7_: Mn^4+^ (KNF) and K_2_Nb_1−*x*_Mo*_x_*F_7_: Mn^4+^ with *x* = 0.05 (KNMF) are shown in Appendix A, and the corresponding basis for the band gap calculation is also provided in the Appendix A. It can be seen that the two samples have strong absorption bands in both the near ultraviolet area and the blue light area, corresponding to the transitions of ^4^A_2g_→^4^T_1g_ and ^4^A_2g_→^4^T_2g_ of Mn^4+^, respectively. More importantly, the absorption band in the blue light area is stronger than the absorption band in the ultraviolet region, proving that the obtained phosphors can be well combined with the blue light InGaN chip. The obtained *E*_g_ value for KNF and KNMF are 4.62 eV and 4.38 eV, respectively. Apparently, the band gap of KNMF obviously decreased after Mo^6+^ being doped in KNF, indicating that the Mo^6+^-doping can effectively improve excitation efficiency of the K_2_NbF_7_: Mn^4+^ phosphors.

Since the charge number of Mo^6+^ is greater than that of Nb^5+^, after a certain amount of Nb^5+^ ions are replaced by Mo^6+^ ions, the crystal field strength of the [MoF_7_]^−^ complexation ion in the formed K_2_Nb_1−*x*_Mo*_x_*F_7_: Mn^4+^ matrix is stronger than that of the [NbF_7_]^2−^ complexation ion, thus effectively narrowing the band gap of the K_2_NbF_7_: Mn^4+^ phosphor. Therefore, the obtained K_2_Nb_1−*x*_Mo*_x_*F_7_: Mn^4+^ solid solution phosphors have the lower photon energy required in the excitation luminescence process, leading to the higher excitation efficiency and the superior luminescence properties in comparison to K_2_NbF_7_: Mn^4+^.

To comprehensively understand the luminescent properties of the as-obtained phosphors, the quantum yields (*η*) of the KNF and KNMF were measured. On the basis of the PL spectra of the two samples, their quantum efficiencies (*η*) are calculated through the following formula:(4)$$\eta =\frac{\int L_{s}}{\int E_{R}-\int E_{S}}$$
where *L*_S_, *E*_S_ and *E*_R_ are the emission spectrum of the phosphor sample, the spectra of excitation light with and without samples in the integral sphere, respectively [51], the quantum yield (*η*) of KNF and KNMF are calculated to be 39.65% and 47.22%, respectively (Appendix A). After doping a suitable concentration of Mo^6+^ ions, the quantum yield of K_2_Nb_1−*x*_Mo*_x_*F_7_: Mn^4+^ solid solution phosphors also can be effectively enhanced compared with the K_2_NbF_7_: Mn^4+^ phosphor.

The decay curves of K_2_Nb_1−*x*_Mo*_x_*F_7_: Mn^4+^ (*x* = 0, 0.05 and 0.15) under excitation of 470 nm blue light (monitored at 627 nm) are depicted in Figure 3d. All the decay curves can be well fitted in a double-exponential function below.
(5)$$I_{\left(t\right)}=I_{0}+A_{1}e^{\left(-t/\tau_{1}\right)}+A_{2}{}^{\left(-t/\tau_{2}\right)}$$
(6)$$\tau =\frac{\left(A_{1}\tau_{1}^{2}+A_{2}\tau_{2}^{2}\right)}{\left(A_{1}\tau_{1}+A_{2}\tau_{2}\right)}$$
where A_1_ and A_2_ are constant (represent the initial intensities of the *τ*_1_ and *τ*_2_, i.e., their relative weightings), *I*_t_ is the luminescence intensity at time t, *I*_0_ is the luminescence intensity at initial time, *τ*_1_ and *τ*_2_ represent the decay lifetime, and *τ* is the average decay time. Among them, *τ*_1_ is short lifetime, while *τ*_2_ is long lifetime. For the decay model by double-exponential fitting, the obtained fast and slow lifetimes can be understood as follows: Assuming that the long lifetime describes the intrinsic emission decay of all Mn^4+^ ions while the short lifetime represents the part of Mn^4+^ ions that have some defect centers nearby, and a corresponding efficient channel of energy transfer to quenching centers. The decay curves deviate from the single exponential model, indicating that an extra energy decay channel is active. This extra energy decay channel may have stemmed from the cation (such as K^+^) vacancy defects in the vicinity of Mn^4+^ ions for charge compensation. The fitting parameters of decay lifetimes for the K_2_Nb_1−*x*_Mo*_x_*F_7_: Mn^4+^ phosphors are provided in Appendix A. The calculated lifetime for the K_2_Nb_1−*x*_Mo*_x_*F_7_: Mn^4+^ phosphors with *x* = 0, 0.05 and 0.15 is 2.1459 ms, 3.4917 ms and 3.1883 ms, respectively. Obviously, the decay lifetime of K_2_Nb_1−*x*_Mo*_x_*F_7_: Mn^4+^ phosphors are significantly larger than that of the K_2_NbF_7_: Mn^4+^ phosphor, further suggesting that Mo^6+^-doping effectively boosted the PL emission intensity of the K_2_Nb_1−*x*_Mo*_x_*F_7_: Mn^4+^ phosphors. Accordingly, the K_2_Nb_1−*x*_Mo*_x_*F_7_: Mn^4+^ phosphor (*x* = 0.05) exhibits the longest decay lifetime could be attributed the optimum Mo^6+^-doping concentration in the K_2_Nb_1−*x*_Mo*_x_*F_7_: Mn^4+^ solid solution in our case.

The corresponding Commission Internationale de L’Eclairage (CIE) coordinates of the K_2_Nb_1−*x*_Mo*_x_*F_7_: Mn^4+^ phosphors are shown in Figure 3c and Table 3, the color purity and the correlated color temperature (CCT) of these phosphors were calculated (the relevant calculation basis and details are provided in Appendix A), which are also listed in Table 3. The coordinates of these phosphors all fall in the redlight region and almost overlap, which are close to the coordinates specified by the National Television Standards Committee (NTSC) (*x* = 0.67, *y* = 0.33), manifesting again that K_2_Nb_1−*x*_Mo*_x_*F_7_: Mn^4+^ samples are superior red phosphors. It is worth mentioning that the obtained K_2_Nb_1−*x*_Mo*_x_*F_7_: Mn^4+^ samples have the higher color purity ~94% (above 90%) and the lower CCT of ~3500 K (below 4500 K), exhibiting the great application potential in warm WLEDs for indoor lighting.

Normally, WLEDs need to work at temperature much higher than room temperature, so the thermal stability of the phosphors is an important parameter to evaluate their actual application. As an example, the thermal stability of the K_2_Nb_1−*x*_Mo*_x_*F_7_: Mn^4+^ with *x* = 0.05 (KNMF) was assessed, and the results are shown in Figure 4. Figure 4a depicts the temperature-dependent emission spectra of KNMF phosphor in the test temperature range of 303–423 K. It is clearly observed that the PL emission intensity of the phosphor gradually decreases with the increasing test temperature, this should be ascribed the thermal quenching effect stemming from the nonradiative transition [42,52]. Whereas the PL patterns do not vary with the increasing temperature, suggests that the luminescence mechanism of the K_2_Nb_1−*x*_Mo*_x_*F_7_: Mn^4+^ phosphors has not changed in the test temperature range. Generally, the ratio of luminescence intensity of a phosphor at the test temperature of 353K (80 °C) to that at the initial temperature is used for evaluating its thermal stability. Figure 4b shows the normalized emission intensity of the red emission peak at 627 nm as the function of temperature for KNMF. Excitingly, the luminescence intensity of the KNMF retains 69.95% of the initial intensity at 353 K, showing the good thermal stability. For a better understanding of the thermal quenching characteristics for the K_2_Nb_1−*x*_Mo*_x_*F_7_: Mn^4+^ (*x* = 0.05) (KNMF) phosphor, the activation energy E_a_ is estimated (the details are provided in Appendix A) and given in Figure 4c. The calculated E_a_ is 0.74 eV, further verifying that the KNMF phosphor owns good thermal stability.

Table 4 also gives comparison of the thermal stability between K_2_Nb_1−*x*_Mo*_x_*F_7_: Mn^4+^ (*x* = 0.05) (KNMF) phosphor with the reported some typical Mn^4+^-activated fluoride phosphors. Comparatively speaking, the as-obtained phosphor shows good thermal stability. The above results may be due to the formation of K_2_Nb_1−*x*_Mo*_x_*F_7_: Mn^4+^ solid solution caused by Mo^6+^-doping and the simultaneous change of the crystal field local environment of Mn^4+^, which not only increases the structural rigidity of a phosphor, but also inhibits the non-radiation transition, thus effectively improving the thermal stability of the phosphors. Figure 4d displays the energy conversion process and mechanism of the thermal quenching of Mn^4+^ in K_2_Nb_1−*x*_Mo*_x_*F_7_, which can better explain the thermal quenching effect. Under excitation of 470 nm blue light, the electrons of Mn^4+^ transition from the ground state ^4^A_2g_ to the excited state ^4^T_1g_ and ^4^T_2g_. These unstable electrons will then make a non-radiative transition to a lower intermediate state ^2^E_g_. Afterwards, the electrons at ^2^E_g_ will return to their ground state by a radiative transition, emitting red light. Nevertheless, some electrons in ^2^E_g_ will assimilate the activation energy under the high temperature, return to the ground state through the intersection of ^4^T_2g_ and ^4^A_2g_. This process is a non-radiative transition, reducing the possibility of normal radiative transition and resulting in the weakening of the intensity of the red emission of Mn^4+^, thus generating thermal quenching behavior.

### 2.3. Moisture Resistance Properties

As is well known, the practical application of Mn^4+^ activated fluoride red phosphors are greatly limited by their poor moisture resistance, thus the moisture resistance is an important performance parameter that should be considered for this kind of phosphors. The moisture resistance properties of K_2_NbF_7_: Mn^4+^ (KNF) and K_2_Nb_1−*x*_Mo*_x_*F_7_: Mn^4+^ with *x* = 0.05 (KNMF) phosphors were measured and shown in Figure 5. The two phosphors were measured based on an extreme moisture resistance method–water immersing [20]. After immersing for various times (0–1440 min), the photographs of the KNF and KNMF red phosphors under the sunlight and blue light irradiation are revealed in Figure 5a,b, respectively, and normalized intensity of the strongest peak at 627 nm of the two phosphors are shown in Figure 5c. Under irradiation of the sunlight, the color of the KNF solution turned into dark purple after being immersed in deionized water for 1440 min, suggests that [MnF_6_]^2−^ ions in KNF phosphor have been obviously hydrolyzed. In contrast, the KNMF solution is almost colorless under the same immersing time (1440 min), implying the superior water stability of [MnF_6_]^2−^ ions in KNMF phosphor (Figure 5a). Moreover, the KNMF phosphor displays a stronger red light than that of the KNF phosphor under the excitation of blue light at 470 nm, intuitively confirming that the better moisture resistance of KNMF phosphors (Figure 5b). It can be seen that from Figure 5c, the red emission intensity (627 nm) of the two phosphors both gradually decrease with the increase of the immersing time, while the decline trend of the red emission intensity for the KNMF phosphor is obviously slower than that for KNF phosphor. More importantly, the red emission intensity of the KNMF phosphor can still maintain 86.37% of its initial level even after immersing for 1440 min, obviously higher than that of KNF phosphor at the same immersing time. These results confirmed that the moisture resistance of K_2_NbF_7_: Mn^4+^ has been obviously enhanced after doping a proper amount of Mo^6+^ ions. Compared with the previously reported Mn^4+^-activated fluorides, the K_2_Nb_1−*x*_Mo*_x_*F_7_: Mn^4+^ (*x* = 0.05) (KNMF) phosphor show excellent water-proof properties (Table 5). The reason is that the formation of solid solutions inhibits the hydrolysis of [MnF_6_]^2−^, thus enhances the moisture resistance of the Mn^4+^-activated fluoride phosphors [43].

### 2.4. Performances of WLEDs

Given that K_2_Nb_1−*x*_Mo*_x_*F_7_: Mn^4+^ phosphors exhibit excellent emission efficiency, good thermal stability, and superior moisture resistance, it is essential to assess their practical application in WLEDs. Three representative WLEDs were packaged: a blue chip (InGaN) with a yellow YAG: Ce^3+^ phosphor (A); a blue chip with a mixtures of YAG: Ce^3+^ and KNF (B); and a blue chip with a mixtures of YAG: Ce^3+^ and KNMF (C). The electroluminescence (EL) spectra (the driving current of 0.7A) and photographs of the WLEDs are exhibited in Figure 6a. It can be seen that the obtained two red phosphors (KNF and KNMF) are efficiently excited by the blue LED chip, resulting in a sharp emission peak in the range of 600–650 nm. In addition, the two WLEDs (B and C) packaged with a blue chip and mixtures of YAG: Ce^3+^ and the K_2_Nb_1−*x*_Mo*_x_*F_7_: Mn^4+^ (*x* = 0 and 0.05) red phosphors display warm white light, while the WLED packaged with a blue chip (InGaN) and yellow YAG: Ce^3+^ phosphor shows cold white light (insets in Figure 6a). The corresponding chromaticity coordinates of these WLEDs are displayed in Figure 6b. Among them, the WLED packaged with a blue chip (InGaN) and yellow YAG: Ce^3+^ phosphor has the CIE color coordinates of (0.3174, 0.3463), a CCT of 6170 K, and the color rendering index (CRI) of 72, indicating that it is not suitable for indoor lighting due to the absence of a red-light component. After adding KNF phosphor as the red component, the packaged WLED possesses the CIE color coordinates of (0.3510, 0.3767), a high CRI of 83.7, and a low CCT of 4883 K. It is worth mentioning that the packaged WLED incorporating KNMF red phosphor has the CIE color coordinates of (0.3886, 0.4035), a higher CRI of 88, and a lower CCT of 3979 K. Compared with currently commercially used WLEDs, the obvious enhancement in CRI and the significant decrease in CCT for the packaged WLED using the synthesized K_2_Nb_1−*x*_Mo*_x_*F_7_: Mn^4+^ phosphors as the red component can be simultaneously achieved. These results further confirm that the as-obtained K_2_Nb_1−*x*_Mo*_x_*F_7_: Mn^4+^ phosphors are promising luminescent materials with potential application in warm WLEDs.

## 3. Materials and Methods

### 3.1. Materials

The raw materials KMnO_4_ (99.5%), HF (40 wt%), H_2_O_2_, and anhydrous ethanol were obtained from the Chengdu Kelon Chemical Reagent Factory. KHF_2_ (99%), Nb_2_O_5_ (99.5%), and H_24_Mo_7_N_6_O_24_·4H_2_O (99%) were purchased from Aladdin Biotechnology Co., Ltd. (Xi’an, China). H_2_O_2_ and anhydrous ethanol were analytical grade and required no further purification.

### 3.2. Synthesis of K_2_MnF_6_ Precursor

Firstly, 10 g of KMnO_4_ and 5 g of KHF_2_ were dissolved in 100 mL of HF (40 wt%) solution. Afterwards, H_2_O_2_ was slowly added drop by drop after vigorously stirring and dissolving for 30 min. The color of the solution gradually changed from dark purple to brownish red, and the golden precipitation of K_2_MnF_6_ was produced. Ultimately, the K_2_MnF_6_ powders were obtained after centrifuging at 4000 r/min for 1min, washing with anhydrous ethanol three times, and drying at 70 °C for 8 h.

### 3.3. Preparation of K_2_Nb_1−x_Mo_x_F_7_: Mn^4+^

In a typical synthesis procedure, Nb_2_O_5_ and H_24_Mo_7_N_6_O_24_·4H_2_O with a preset stoichiometric ratio (keeping the total amount of Nb and Mo of 1.9 mmol) and 6 mL HF were added into the Teflon-lined autoclave. Subsequently, the autoclave was maintained at 120 °C for 30 min. After the autoclave was naturally cooled to room temperature, 0.1 mmol of the synthesized K_2_MnF_6_ was added into the resultant solution and vigorously stirred for 10 min. Then 4 mmol of KHF_2_ was put into the as-obtained solution and continuously stirred for 10 min. After that, 5mL ethanol was added to the above solution and stirred thoroughly for 20 min until the reaction stopped. The sediments were collected, washed with anhydrous ethanol three times, and dried at 80 °C for 12 h. Lastly, the K_2_Nb_1−*x*_Mo*_x_*F_7_: Mn^4+^ red phosphors were obtained (Appendix A).

### 3.4. Characterization

The crystal structures of the as-synthesized products were characterized with an Xray powder diffractometer (Cu Kα radiation, λ = 1.5406 Å). The structure was optimized by the GSAS program. The morphologies and the actual doping concentration of Mn^4+^ in a solid solution crystal were obtained with a scanning electron microscope (SEM, Quanta 250, FEI, St. Leonards, NSW, Australia) and Agilent 8900 ICP-MS (Santa Clara, CA, USA), respectively. Photoluminescence spectra (PL), photoluminescence excitation spectra (PLE), and temperature-dependent luminescence properties were tested employing a Hitachi F-4600 Fluorescence analyzer (Tokyo, Japan) equipped with a 150 W Xe lamp as an excitation light source. The diffuse reflection spectra of the samples were tested with a spectrophotometer (UV-3600, Shimadzu, Tokyo, Japan) and BaSO_4_ was used as the reflective material. The quantum efficiency (QE) of the samples was measured by a fluorescence spectrometer with an integrating sphere (Horiba FluoroMax-4, Tokyo, Japan). The WLEDs were tested by a high-precision array spectrometer (HSP 6000) under a steady current of 700 mA.

## 4. Conclusions

Novel Mn^4+^-activated K_2_Nb_1−*x*_Mo*_x_*F_7_ phosphors were designed and successfully synthesized through a co-precipitation method. Benefiting from the synergistic effect of “solid solution formation” and “charge compensation” induced by Mo^6+^ doping, the obtained K_2_Nb_1−*x*_Mo*_x_*F_7_ phosphors exhibit obviously enhanced comprehensive luminescence properties, including PL intensity, quantum yield, thermal stability, and moisture resistance. Notably, the obtained K_2_Nb_1−*x*_Mo*_x_*F_7_: Mn^4+^ (*x* = 5%) phosphor exhibits the optimal comprehensive luminescence properties among the K_2_Nb_1−*x*_Mo*_x_*F_7_: Mn^4+^ series. Using the K_2_Nb_1−*x*_Mo*_x_*F_7_: Mn^4+^ (*x* = 5%) phosphor as a red component, the packaged WLED possesses a high CRI of 88 and a low CCT of 3979 K. The as-developed K_2_Nb_1−*x*_Mo*_x_*F_7_: Mn^4+^ phosphors significantly boost the CRI (from 72 to 88) and decrease the CCT (from 6170K to 3979K) when compared with the commercial WLED packaged with a blue chip (InGaN) and yellow YAG: Ce^3+^ phosphor, with attractive application prospectives in WLEDs for indoor lighting.

## Figures and Tables

**Figure 1 molecules-28-04566-f001:**
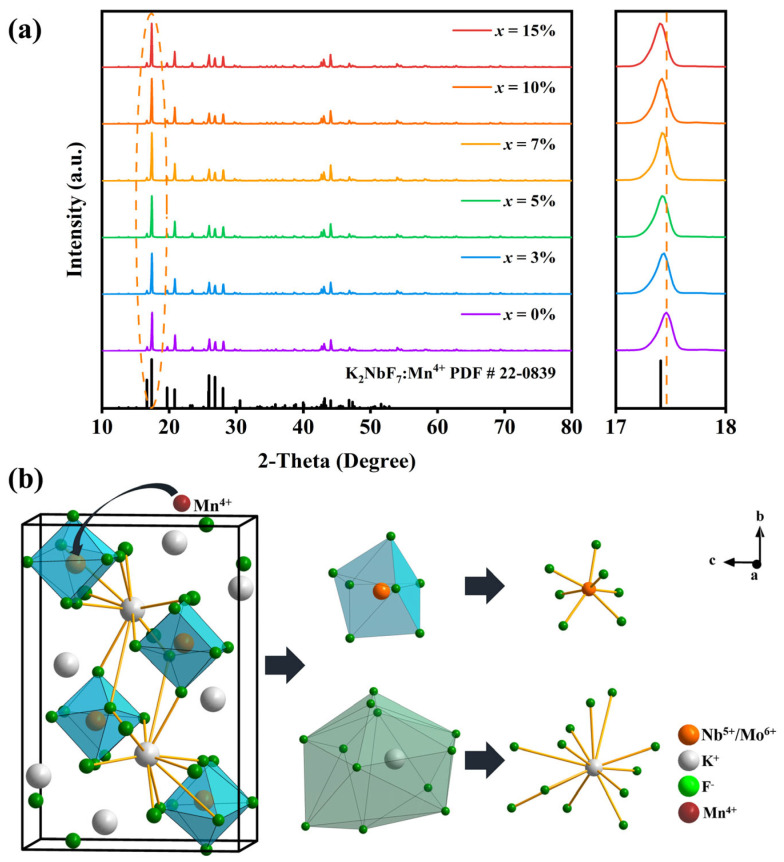
(**a**) XRD patterns of K_2_Nb_1−*x*_Mo*_x_*F_7_: Mn^4+^ phosphors, (**b**) crystal structure of K_2_Nb_1−*x*_Mo*_x_*F_7_: Mn^4+^ solid solution.

**Figure 2 molecules-28-04566-f002:**
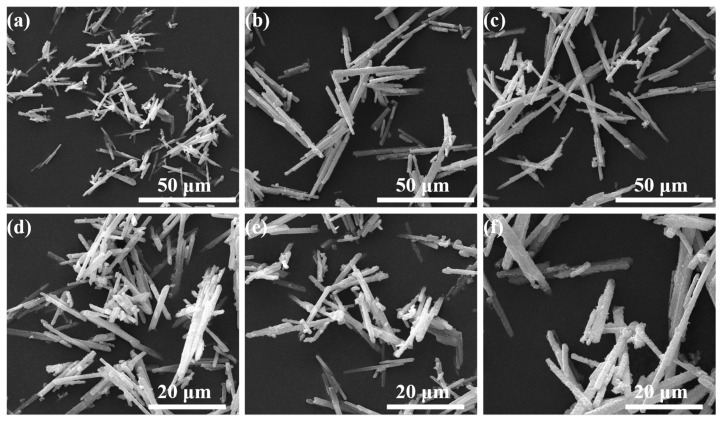
SEM images of the representative K_2_Nb_1−*x*_Mo*_x_*F_7_: Mn^4+^ (*x* = 0, 0.05, 0.15) samples. (**a**,**d**): *x* = 0; (**b**,**e**): *x* = 0.05; (**c**,**f**): *x* = 0.15.

**Figure 3 molecules-28-04566-f003:**
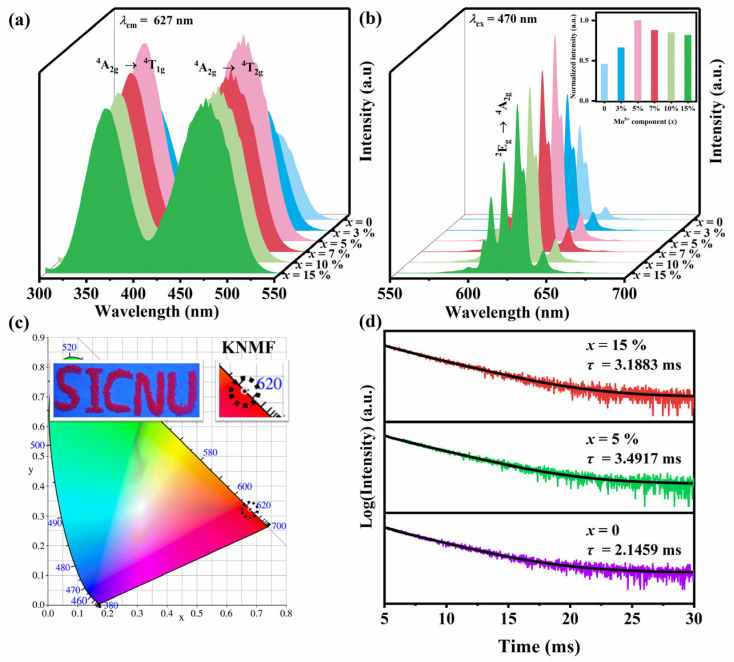
Excitation spectra (**a**), emission spectra (**b**), CIE chromaticity coordinates (**c**), and decay curves (**d**) of K_2_Nb_1−*x*_Mo*_x_*F_7_: Mn^4+^ phosphors.

**Figure 4 molecules-28-04566-f004:**
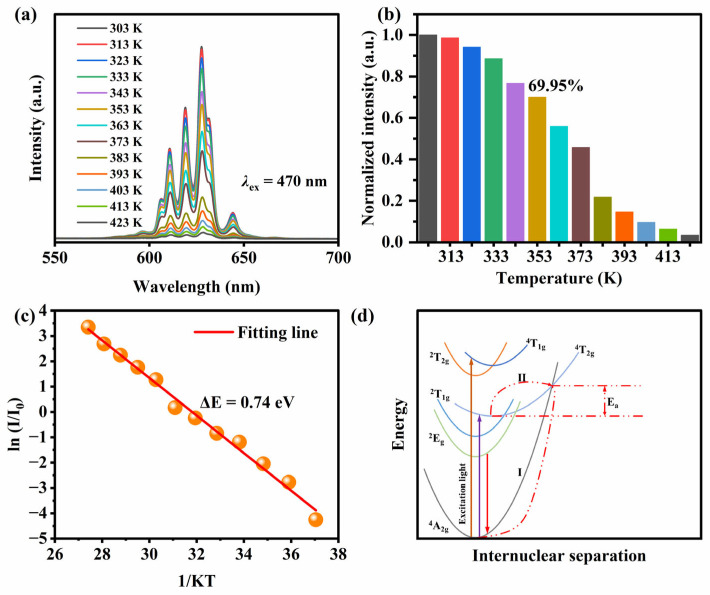
Temperature-dependent emission spectra of K_2_Nb_1−*x*_Mo*_x_*F_7_: Mn^4+^ with *x* = 5% (KNMF) (**a**); Normalized emission intensity of the red emission peak at 627 nm as the function of temperature for KNMF (**b**); Plot of ln(I_0_/I-1) versus 1/kT for KNMF (**c**); Configurational coordinate diagram of Mn^4+^ in the K_2_Nb_1−*x*_Mo*_x_*F_7_: Mn^4+^ phosphors (**d**).

**Figure 5 molecules-28-04566-f005:**
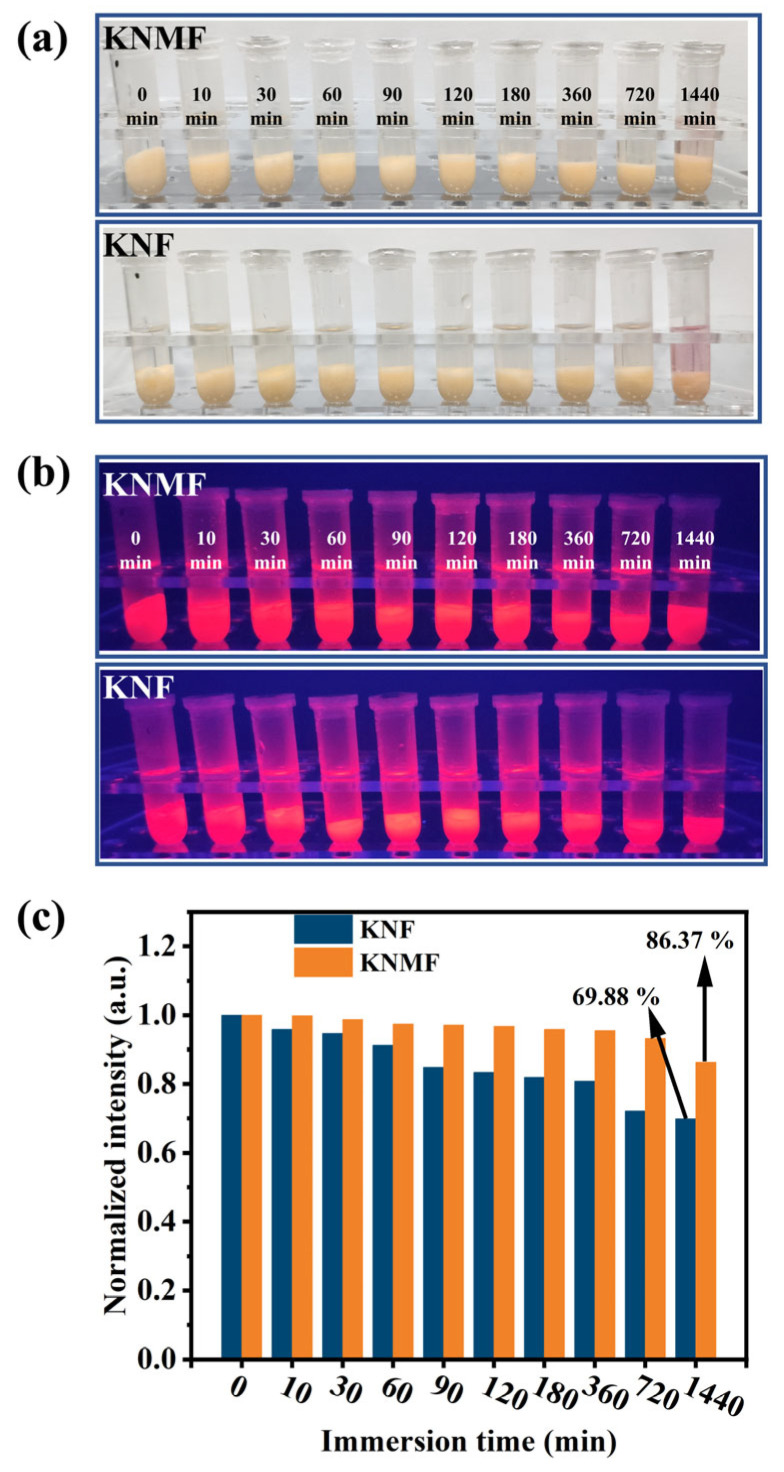
Photographs of K_2_NbF_7_: Mn^4+^ (KNF) and K_2_Nb_1−*x*_Mo*_x_*F_7_: Mn^4+^ with *x* = 0.05 (KNMF) phosphors after immersing into deionized water for different times under the sunlight (**a**) and blue light excitation at 470 nm (**b**). The normalized intensity of the strongest peak at 627 nm of the KNF and KNMF phosphors as a function of immersion time (**c**).

**Figure 6 molecules-28-04566-f006:**
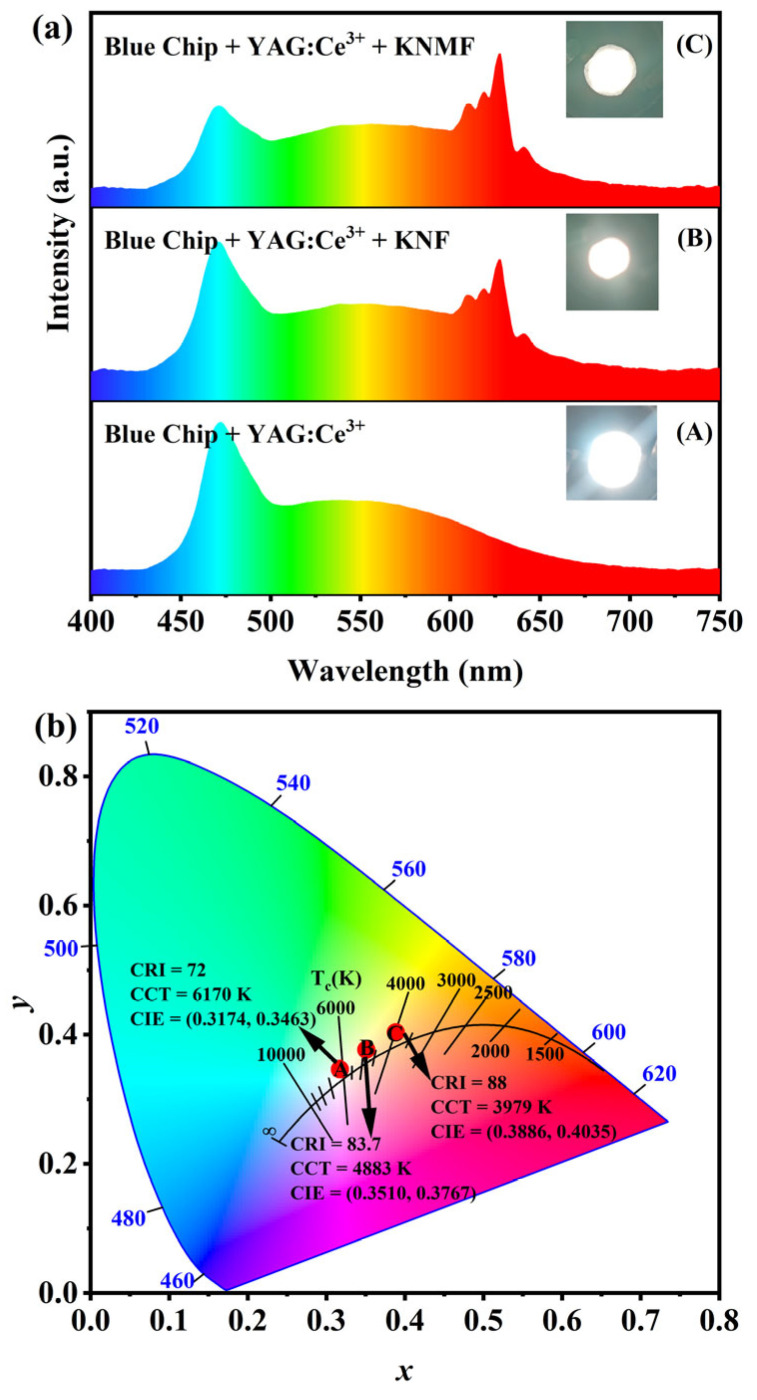
Electroluminescence spectra (**a**) and CIE chromaticity diagram (**b**) of the WLEDs were packaged by employing a blue chip (InGaN) with yellow YAG: Ce^3+^ phosphor—A, blue chip (InGaN) with the mixtures of YAG: Ce^3+^ and KNF—B, blue chip (InGaN) with the mixtures of YAG: Ce^3+^ and KNMF—C. Insets show the photographs of the fabricated WLEDs.

**Table 1 molecules-28-04566-t001:** XRD refinement results and lattice parameters of the K_2_Nb_1−*x*_Mo*_x_*F_7_: Mn^4+^ (0≤ *x* ≤ 0.15) crystals.

K_2_Nb_1−*x*_Mo*_x_*F_7_: Mn^4+^	*x* = 0	*x* = 3%	*x* = 5%	*x* = 7%	*x* = 10%	*x* = 15%
*R_wp_* (%)	7.10	7.33	6.95	7.58	6.83	5.47
*R_p_* (%)	5.01	5.23	5.01	5.27	5.10	4.10
*χ* ^2^	3.612	3.889	3.537	4.360	3.429	2.193
*a* (Å)	5.8452	5.8466	5.8469	5.8480	5.8476	5.8470
*b* (Å)	12.6922	12.6913	12.6910	12.6911	12.6924	12.6947
*c* (Å)	8.5138	8.5134	8.5138	8.5138	8.5144	8.5156
Cell volume (Å^3^)	631.624	631.697	631.752	631.871	631.930	632.067

**Table 2 molecules-28-04566-t002:** The actual doping concentrations of Mo^6+^ ions and Mn^4+^ ions in the K_2_Nb_1−*x*_Mo*_x_*F_7_: Mn^4+^ crystals.

The Designed K_2_Nb_1−*x*_Mo*_x_*F_7_: Mn^4+^ Samples	Actual Molar Ratio of Mo^6+^ in K_2_Nb_1−*x*_Mo*_x_*F_7_: Mn^4+^ Crystals (%)	Actual Molar Ratio of Mn^4+^ in K_2_Nb_1−*x*_Mo*_x_*F_7_: Mn^4+^ Crystals (%)
*x* = 0	0	1.21
*x* = 3%	0.24	1.19
*x* = 5%	0.39	1.19
*x* = 7%	0.91	1.21
*x* = 10%	1.05	1.24
*x* = 15%	1.46	1.48

**Table 3 molecules-28-04566-t003:** CIE chrominance coordinates and CCT values of the K_2_Nb_1−*x*_Mo*_x_*F_7_: Mn^4+^ phosphors.

K_2_Nb_1−*x*_Mo*_x_*F_7_: Mn^4+^	CIE (*x, y*)	Color Purity (%)	CCT (K)
*x* = 0	(0.6759, 0.3223)	93.60	3533
*x* = 3%	(0.6755, 0.3226)	93.50	3513
*x* = 5%	(0.6779, 0.3206)	94.11	3645
*x* = 7%	(0.6770, 0.3214)	93.88	3592
*x* = 10%	(0.6771, 0.3213)	93.90	3599
*x* = 15%	(0.6775, 0.3210)	94.00	3619

**Table 4 molecules-28-04566-t004:** Comparison of the thermal stability between K_2_Nb_1−*x*_Mo*_x_*F_7_: Mn^4+^ (*x* = 0.05) (KNMF) phosphor with the reported some typical Mn^4+^-activated fluoride phosphors.

Samples	PL Normalized Intensity	Activation Energy (eV)	Ref.
K_2_Nb_1−*x*_Mo*_x_*F_7_: Mn^4+^ (*x* = 0.05)	69.95%@353 K	0.74	This work
K_2_NbF_7_: Mn^4+^	70%@348 K	0.66	[4]
K_2_TiF_6_: Mn^4+^	/	0.70	[4]
K_2_LiAlF_6_: Mn^4+^	/	0.62	[4]
K_2_NaGaF_6_: Mn^4+^	76%@398 K	/	[17]
BaTiF_6_: Mn^4+^	44%@425 K	0.628	[20]
K_0_._07_Ba_0_._965_TiF_6_: Mn^4+^	60%@425 K	0.940	[20]
Cs_2_KAlF_6_: Mn^4+^	59.8%@423 K	/	[27]
Cs_2_RbAlF_6_: Mn^4+^	72.1%@423 K	/	[27]
K_2_LiAlF_6_: Mn^4+^	51.5%@423 K	/	[27]
(NH_4_)_2_TiF_6_: Mn^4+^	50%@343 K	0.3123	[30]
(NH_4_)_2_SiF_6_: Mn^4+^	64%@323 K	0.4619	[30]
K_2_TaF_7_: Mn^4+^	70.9%@343 K	/	[40]
BaTiF_6_: Mn^4+^	70%@425 K	0.84	[52]

**Table 5 molecules-28-04566-t005:** Comparison of moisture resistance between K_2_Nb_1−*x*_Mo*_x_*F_7_: Mn^4+^ (*x* = 0.05) (KNMF) phosphor with the reported typical Mn^4+^-activated fluoride phosphors.

Samples	Immersion Time	Intensity	Reference
K_2_Nb_1-*x*_Mo*_x_*F_7_: Mn^4+^ (*x* = 0.05)	1440 min	86.37%	This work
K_2_NbF_7_: Mn^4+^	1440 min	69.88%	This work
K_2_TiF_6_: Mn^4+^	120 min	6.8%	[1]
K_2_TiF_6_: Mn^4+^@CaF_2_	120 min	86.4%	[1]
BaGeF_6_: Mn^4+^	7200 min	15%	[10]
BaGeF_6_: Mn^4+^@PPG	7200 min	35%	[10]
K_3_RbGe_2_F_12_: Mn^4+^	600 min	40%	[13]
K_2_NaGaF_6_: Mn^4+^(2.04 at%)	420 min	83%	[17]
K_2_SiF_6_: Mn^4+^(3.31 at%)	420 min	23%	[17]
K_2_TiF_6_: Mn^4+^	150min	20%	[20]
BaTiF_6_: Mn^4+^	150 min	50%	[20]
K_0.07_Ba_0.965_TiF_6_: Mn^4+^	150 min	65%	[20]
Cs_2_KAlF_6_: Mn^4+^	4320 min	65%	[27]
Na_2_SiF_6_: 0.06Mn^4+^	300 min	32%	[43]
Na_2_GeF_6_: 0.06Mn^4+^	300 min	33%	[43]
Na_2_Si_0.5_Ge_0.5_F_6_: 0.06Mn^4+^	300 min	71%	[43]
Rb_2_SnF_6_: Mn^4+^	30 min	10.3%	[51]
K_3_(NbOF_5_)(HF_2_): Mn^4+^	360 min	74%	[53]

## Data Availability

Not applicable.

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
