# Peer review of "Novel Mn4+-Activated K2Nb1−xMoxF7 (0 ≤ x ≤ 0.15) Solid Solution Red Phosphors with Superior Moisture Resistance and Good Thermal Stability"

_molecules, 2023, doi:10.3390/molecules28114566_

Round 1
Reviewer 1 Report
Overall, several matters posed in the paper need to be addressed:
1) Introduction: The focal objective and highlight of this research work could be further strengthen.
2) Pg 3: labelling of figures in Figure 1 should be as follows: (a) XRD patterns of K2Nb1-xMoxF7: Mn4+ phosphors, (b) crystal structure of K2Nb1-xMoxF7: Mn4+ solid solution
3) Pg4, line 117: recheck this “P121/c1
4) Pg4-Pg6: check grammars and sentence structure. Recommended to use shorter sentences.
5) Pg8, Figure 4b: needs to be improved. The values in x-axis is overcrowded.
6) Pg9-Pg10: Table 4 should be fitted into 1 page instead of 2 pages.
7) Pg11, Table 5: it might be useful if authors changed and convert the immersion time to minutes for ease of comparison.
8) Pg11,Figure 5: the photographs need to be resized in order to have appropriate width and height.
9) Pg10, line 308-312: Confusing sentences leading to confusion in meaning for the text. Authors need to rework this sentence carefully. It is advisable to use shorter sentences.
10) Pg13, Conclusion: Recommended to include how this research work will contribute to the ongoing WLEDs research.
11) In the front page of Supplementary materials: For content, it is recommended to list down all figures and tables for ease of readers.
In conclusion, there are many grammatical errors and incorrect sentence structures. Authors should have their work checked for “English expression, grammars and formatting carefully before submitting a revised version. Some careful proof-reading is needed. I therefore suggest Minor Revision.
There are many grammatical errors and incorrect sentence structures. Authors should have their work checked for “English expression, grammars and formatting carefully before submitting a revised version. Some careful proof-reading is needed.
Reviewer 2 Report
In this manuscript, the authors describe the "Novel Mn4+-activated K2Nb1-xMoxF7 (0 ≤ x ≤ 0.15) Solid Solution Red Phosphors with Superior Moisture Resistance and Good Thermal Stability". While this manuscript is generally well written, the authors should address the minor issues below:
1) In line 39, the word should be "interestingly" not "amusedly" as written. The authors should consider revising it.
2) In lines 131 - 132, the sentence needs more clarity. The authors should consider revising it. The sentence would be much clearer if the word "suggests" is replaced with "suggesting". The authors should consider revising that.
3) In line 285, the beginning sentence should be: "As is well known...."
4) To avoid or at least minimize any unnecessary distractions, it would be helpful if figures are placed on the same page as its description. A case in point is Figures 5a, b and c in lines 292 - 294.
5) In line 329, not sure what the authors meant by "availably".... The authors should re-word that sentence.
In summary, this manuscript could potentially benefit its audience if the above issues are adequately addressed.
The English grammar is fine with minor issues.
Reviewer 3 Report
This paper describes the preparation of a Mn4+ activated fluoride red phosphor with superior moisture resistance and good thermal stability. The work seems to be well conducted.
- However, a recent paper with a similar objective was not cited (Dalton Trans., 2021,50, 17290-17300).
- Moreover, the performance of the prepared WLED, using the synthesized red phosphors, should be compared with the currently commercially used WLED and the advantages of the proposed new solution stressed.
- Raw materials sustainability issues should be discussed.
Reviewer 4 Report
The manuscript reports on the synthesis and characterization of red phosphors. The main finding of the work is that Mn-doped KNMF phosphor exhibits improved optical and thermal properties as compared to the KNF counterpart. The manuscript is well organized, the presented work is detailed, and the results may have a positive impact on red phosphor commercialization for WLEDs. With these general remarks in mind, below we are offering our comments to the authors.
The authors have indicated the importance of the charge compensation in Mn-doped KNF and KNMF; however, this issue is not considered in the discussion. The creation of point defects and/or complex defects due to charge compensation is completely omitted. This makes the energy migration, and luminescence kinetics discussion speculative. The authors shall critically address this issue.
Please provide a reference to the GSAS software used.
Table 1 – please present the physical parameters’ numerical values within real world accuracy, avoid artificial numbers with many digits after the decimal point due to the software.
Please show SEM images (Fig.2) with higher magnification. We believe that the rod-like microcrystal surface is not smooth, thus a new argument may be considered when discussing luminescence quenching.
Why in Table 2, the molar ratio for KNMF with x=0 shows the Mn content of 1.21 at.%. Was this fact considered when calibrating for remaining x values?
Was the KNMF rod-like microcrystal powder powdered (crushed to make uniform particles distribution) before using it for mixing with the yellow phosphor for WLED tests? If not, then how does the change of surface-to-volume ratio change in powdered phosphor (it is required for achieving high compactness of phosphor used in WLEDs) will affect the luminescence efficiency? The authors shall critically address this issue.
PL kinetics analysis omitted a fast decay component in addition to claimed two exponential decay components. What is the PL excitation pulse decay curve (decay time) used for PL kinetics? Were the PL kinetics days corrected for the excitation pulse decay? Present Fig.3(d) Y-axes using a logarithmic scale.
Please identify the physical meaning of each exponential decay component identified.
Was the thermal stability of the phosphor-converted LEDs experiment done using a close-proximity phosphor arrangement? The information shall be included in the experimental section.
Does phonons’ energy change significantly between KNF and KNMF? Does it change with Mn doping?
Please discuss the phonons’ involvement in energy migration cooperative processes.
Were phosphors used in moisture resistance experiments in dry or wet form?
Please extrapolate the results of the moisture resistance experiments results beyond 24 hrs, to be on a typical time scale with a WLEDs lifetime (~10,000 hrs). What would be the intensity reduction then?
In conclusion, the results presented are interesting and promising; however, the data interpretation and discussion are very optimistic and do not address many critical issues as indicated, which BTW may change the overall assessment of the new synthesized Mn-doped KNMF red phosphor performance parameters. We recommend a major revision before making final recommendations.
| Minor editing of English language required |
Round 2
Reviewer 3 Report
The authors have satisfactorily answered to the referee questions.
Reviewer 4 Report
The authors have implemented all suggested changes and answered the reviewer’s questions satisfactorily. Thank you!
The manuscript requires one more round of English proofreading to eliminate existing minor flaws (e.g. fermi shall be Fermi).
The manuscript requires one more round of English proofreading to eliminate existing minor flaws (e.g. fermi shall be Fermi).